# A Narrative Review of the Strengths and Limitations of Real-World Evidence in Comparison to Randomized Clinical Trials: What Are the Opportunities in Thoracic Oncology for Real-World Evidence to Shine?

**DOI:** 10.3390/curroncol32110629

**Published:** 2025-11-10

**Authors:** Peter M. Ellis, Larissa Long, Courtney H. Coschi, Arani Sathiyapalan

**Affiliations:** 1Department Oncology, Division of Medical Oncology, Juravinski Cancer Centre, McMaster University, Hamilton, ON L8S 4L8, Canada; coschic@hhsc.ca (C.H.C.); sathiyapal@hhsc.ca (A.S.); 2Faculty of Health Sciences, McMaster University, Hamilton, ON L8S 4L8, Canada; longl8@mcmaster.ca

**Keywords:** real world evidence, strengths, limitations

## Abstract

Randomized clinical trials (RCTs) are considered the gold standard for the evaluation of new interventions and therapies. However, questions have been raised about the generalizability of findings from RCTs. Real-world evidence and real-world data have gained increasing attention as an alternative to data from RCTs. We undertook a review to examine the strengths and limitations of real-world evidence. Real-world evidence may improve the generalizability of clinical trial findings and the assessment of efficacy in under-represented populations. However, real-world evidence studies have poorer internal validity, are unable to adequately adjust for confounding, and have inherent biases in study design. Real-world data may complement existing data from RCTs and play an important role in evaluating patterns and outcomes of care.

## 1. Introduction

Randomized clinical trials are considered the gold standard for the evaluation of new interventions and therapies [1,2]. The process of randomization and stratification provides the best means of balancing known and unknown confounding variables between study groups and is considered the study design that provides the least biased estimate of efficacy [3]. Large randomized trials, or meta-analyses of randomized clinical trials, provide the highest levels of evidence of therapeutic efficacy [4,5]. The results from randomized clinical trials are highly influential in treatment decision-making and decisions about the implementation of new therapeutic options within the field of thoracic oncology.

It is estimated, though, that fewer than 10% of cancer patients receive treatment as part of a clinical trial [6]. While randomized clinical trials are considered to have high internal validity, questions have been raised about the external validity, or generalizability of findings from randomized trials [7]. There are often extensive inclusion and exclusion criteria, designed to maximize the scientific rigor of a trial. However, this process often excludes many patients with lung cancer that have poorer functional status or comorbid health problems and yet are routinely considered for treatment. Therefore, concerns exist about the ability to generalize findings on the effectiveness of new interventions and therapies to the broader population of patients that receive systemic anti-cancer therapies.

In recent years, there has been increasing interest in collecting and reporting information about the effectiveness of therapies in the real world [8]. The concept of real-world evidence (RWE), or real-world data (RWD), has gained increasing attention as an alternative, or supplement, to data derived from randomized clinical trials. Proponents argue that the data are more applicable and generalizable to the broader population, whereas critics argue that the data are subject to large potential bias and are therefore less informative.

We undertook a narrative review of the literature, to further explore the strengths and limitations of real-world evidence in comparison to randomized clinical trials and to better understand the opportunities for the use of real-world evidence in thoracic malignancies.

## 2. Materials and Methods

A narrative review of the medical literature was conducted looking for studies addressing the strengths and limitations of real-world evidence in comparison to randomized clinical trials. The Healthstar and Medline databases via Ovid were searched, using the following search terms: Randomized Controlled Trials, OR Study Design AND Real World Evidence AND Bias, OR Strength, OR Limitation (the full search strategy is available in Appendix A). Articles were eligible for inclusion if they discussed methods to incorporate real-world evidence into research, including the potential strengths or limitations of the use of real-world evidence, and compared real-world evidence to traditionally accepted forms of evidence (e.g., randomized clinical trials, meta-analyses of randomized trials). All relevant articles were included, not just those pertaining to oncology, or thoracic oncology, given that the methodological issues are not oncology-specific. Articles were excluded if they were in abstract form only, were not published in English, or a full text could not be accessed electronically. This research did not require Research Ethics Board review or approval.

The articles retrieved from the literature search were imported into Covidence. Title and abstract screening, full-text review, and data abstraction were conducted by one author (LL). Quality assessment was not performed, as the articles retrieved generally did not report on methods or empiric data and would all be classified as low-quality. A tool such as the Cochrane risk of bias [9] could not meaningfully be applied. There were no quantitative data to combine or analyze. Therefore, information on the strengths and limitations of real-world evidence were abstracted from the articles and analyzed qualitatively. Similar issues were grouped together, and a frequency count was performed to assess how frequently each strength or limitation was reported in the literature. A frequency count was used to indicate how often a strength or limitation was raised in the literature, as a surrogate of the relative importance of the issue. These were then ordered in descending frequency.

## 3. Results

A total of 398 papers were identified from the literature search. After the elimination of duplicates, 233 papers were identified for title and abstract screening, 89 articles were retrieved for full-text screening, and 22 articles were included in the review. Given the nature of the topic, these articles did not contain empirical data but were descriptive in nature. The findings have been summarized as follows:

### 3.1. Strengths of Real-World Evidence

Multiple strengths (Table 1) of real-world evidence were identified from the review of the literature. The most frequently reported strength (n = 12) is that it provides an assessment of the generalizability of findings from randomized clinical trials in the real world. These included the concepts that real-world evidence may be used as an adjunct to randomized trials to determine if their findings translate into improved outcomes in clinical practice, and that the findings may be informative in understanding cost-effectiveness. This is one of the key criticisms of randomized clinical trials. The inclusion and exclusion criteria can severely limit the number of patients that may be eligible for study entry. Excluding patients with poor functional status and significant comorbidity creates a knowledge gap and raises questions on whether clinical trial findings can be extrapolated to these patients. Data from real-world evidence studies may help to fill these knowledge gaps.

Clinical trial findings are often reported with relatively short follow-up. Real-world evidence can provide long-term surveillance data (n = 10), as it is often derived from databases such as electronic health records that provide longer-term follow-up. This can provide additional information regarding the side effects of a therapy and also the long-term treatment outcomes.

There are situations where randomized clinical trials are not feasible to conduct, such as in rare diseases or conditions where randomization is not possible (n = 9). Additionally, there are circumstances such as uncommon molecular subtypes of non-small-cell lung cancer (NSCLC), where there are no or limited randomized data available (n = 7). Data from real-world evidence can provide useful information about treatment efficacy in all of these situations. This is an important role to recognize. Ignoring real-world data in these circumstances can result in missed treatment opportunities for patients. There is a need to clearly define circumstances in which, despite the limitations described below, real-world data will be accepted from regulatory and funding authorities to ensure that these patient groups have access to potentially beneficial therapies.

Many articles identify that real-world data can provide more information on the generalizability of clinical trial findings (n = 7) or treatment patterns under real-world conditions (n = 8), in particular for patient groups that are under-represented in clinical trial populations. Oncology trials generally exclude patients with poorer functional status (ECOG performance status of 2 or higher) and/or significant comorbid health problems. However, these patients often receive treatment in real-world settings. Data from real-world studies can provide information on the outcomes of care, as well as the profile of side effects in these under-represented groups. Studies evaluating real-world patterns of care are also informative about the extent of the uptake of treatment options in a more general patient population and can aid in the identification of gaps in the implementation of evidence into routine care.

Real-world evidence studies are considered to be more efficient in time (n = 7) and resources (n = 9). They may also provide larger sample sizes (n = 7) due to the use of data from electronic medical records, health maintenance organizations, cancer registry data, or provincial/state healthcare databases in publicly funded healthcare systems. These are prospectively collected but generally analyzed retrospectively.

Real-world studies can be used to improve our understanding of prognostic variables or cohorts for stratification in future clinical trials (n = 4). They can also generate hypotheses for future testing (n = 3). Important advances have been made in the field of oncology, as a consequence of astute observations made from real-world clinical data. One such example was the discovery that mutations of the *epidermal growth factor receptor* (*EGFR*) gene-predicted response to EGFR tyrosine kinase inhibitors [31].

### 3.2. Limitations of Real-World Evidence

Almost all of the studies (n = 19) identified the risk of biased data as a major limitation of real-world evidence (Table 2). Real-world data is subject to multiple sources of bias including selection bias, loss to follow-up, the use of historical controls, or the lack of a control group. This is in part due to study designs without adequate control groups, as well as the lack of standardized methodologies for real-world evidence studies. Issues of selection bias, missing data, loss to follow-up, as well as other sources of bias represent the major concern regarding real-world evidence. The balancing of known and unknown confounding variables that are minimized through the process of randomization cannot be easily overcome through other methodologies. Stratification and matching approaches are only able to adjust for major prognostic variables and therefore fail to appropriately address the issue of bias. Real-world studies frequently report on therapeutic effectiveness but fail to take into account the issues of bias. Similarly, lower internal validity and confounding (n = 13) were frequently identified as limitations to real-world evidence. The risk of data quality is also raised in the majority of studies (n = 16). In part, this relates to the previous issues concerning bias. However, real-world data is often extracted from existing databases. Data quality issues can include the retrospective nature of such data, or missing data. Additionally, important variables may not be included in real-world evidence databases (n = 4). For example, in the field of oncology, important variables such as functional status, smoking status, and weight loss may not be present in existing databases.

**Table 2 curroncol-32-00629-t002:** Limitations of real-world evidence.

Limitation	# of Studies Mentioning the Limitation
Risk of biased data [10,11,12,13,14,15,18,19,20,21,22,23,24,25,26,27,29,30,32]	19
Risk of data quality problems [10,13,14,15,16,17,18,19,24,25,26,27,28,29,30,32]	16
Lower internal validity and confounding [11,12,14,20,21,22,23,25,26,27,28,29,32]	13
Inadequate evaluation [12,13,14,17,25,29,32]	7
Lack of randomization [11,14,22,25,29,30]	6
Lack of standardization [14,15,19,28,29]	5
Differences in RWE and RCT due to differences in monitoring frequency/other external circumstances [15,29,30,32]	4
Not inclusive of important clinical information [10,12,25,26]	4
Small sample sizes for rare diseases [14,28,29,32]	4
Cannot hypothesis test (hypothesis-generating) [10,11,29]	3
Lower external validity—convenience sampling [12,15]	2
Risk of inadequate study design [22,25]	2
Conditional on well-designed research question [13]	1
Risk of non-scientific goals [22]	1

A key limitation of real-world evidence pertaining to the effectiveness of interventions is the lack of randomization (n = 6), or lack of standardization (n = 7). Randomization is the most effective way of balancing known and unknown confounding variables. This provides the least biased estimate of the effect of an intervention. The lack of randomization also limits a study’s ability for hypotheses testing and limits real-world evidence for hypothesis generation (n = 3).

Several studies raised a number of issues relating to study design, including the lack of standardization for methodologies for real-world evidence studies (n = 5), differences between real-world evidence and randomized clinical trials in the frequency of monitoring outcomes (n = 4), convenience sampling which can lower external validity (n = 2), the risk of inadequate study design (n = 2), being conditional on a well-designed research question (n = 1), as well as a risk of non-scientific goals (n = 1). All these issues speak to the issue of bias resulting from study design and the risk of this systematically affecting the estimate of the outcomes. Interestingly, four studies identified small sample size as an issue for rare disease. This was also raised as a potential strength, in that real-world evidence might provide some estimate of benefit from an intervention in a scenario where a randomized trial is not possible. This highlights the unique challenges in generating high-quality evidence for rare or uncommon diseases.

## 4. Discussion

Real-world evidence is increasingly reported in the medical literature. In comparison to randomized clinical trials, this review identified many strengths of real-world evidence, including the improvement in the generalizability of clinical trial findings and assessment of therapies in patient populations under-represented in clinical trials. However, the largest and most significant concerns about real-world evidence are reduced internal validity, the inability to adequately adjust for confounding, and the inherent bias in study design. Taken together with the lack of standardization of study design, these all significantly limit the ability of real-world data to assess the effectiveness of therapeutic interventions.

The PACIFIC-R trial [33] was a real-world evidence trial to examine the generalizability of findings from the PACIFIC trial [34]. This was a randomized clinical trial evaluating a one-year treatment course of durvalumab, an immune-checkpoint inhibitor, following concurrent chemoradiation in unresectable stage III NSCLC. PACIFIC demonstrated improvements in both progression-free survival (median PFS 16.9 m vs. 5.6 m, HR 0.55, 95%CI 0.45–0.68) and overall survival (median OS 47.5 m vs. 29.1 m, HR 0.72, 95%CI 0.59–0.89). Patients included in a subsequent worldwide patient access program were enrolled in the PACIFIC-R real-world evidence collection, where possible. The primary outcomes were PFS and OS. The initial data from PACIFIC-R reported a real-world PFS of 21.7 months (95%CI 19.1–24.5 m). Secondary outcomes demonstrate the improved PFS of concurrent versus sequential chemoradiation, patients with PD-L1-positive versus PD-L1-negative tumors, and the use of cisplatin versus carboplatin chemotherapy. The authors conclude that the study confirms the improved efficacy of consolidation durvalumab, but do not take into account patient selection bias and imbalances in prognostic factors as additional explanations for the improved outcomes.

There is a need for additional research to standardize methodologies for real-world data collection [35]. Forethought and attention are required to ensure the sound methodological design of real-world studies: re-labelling research with weak methodologies as real-world evidence will not transform them into methodologically sound study designs. Efforts are needed to better define research questions that can realistically be answered through real-world data collection. Thought needs to be given to defining the study population to limit issues such as selection bias. A greater understanding of the impact from bias resulting from differential outcome assessment and loss to follow-up is needed. Additionally, efforts can be made to statistically match important variables in the analysis. Stratification and adjustment for important prognostic variables need to be considered in analyzing real-world data. This can only be performed, however, on a few important prognostic or confounding variables without risking the introduction of random error from excessive statistical testing. More complicated statistical adjustment techniques, such as propensity matching, may also be considered. However, these still only match important variables, and therefore incompletely adjust for imbalance between groups. Lastly, the authors of real-world data research need to be cognizant of the impact of confounding and bias and not overstate the implications of their research findings.

As a result, randomized clinical trials remain the trial design of choice to assess the benefits of therapeutic interventions. There is a tension between strict inclusion/exclusion criteria and scientific rigor versus the need for improved generalizability of randomized clinical trial findings. The issue, therefore, is what role does real-world evidence play in thoracic malignancies, and how can real-world evidence be used to complement existing data from randomized clinical trials of interventions in thoracic cancers.

### 4.1. Real-World Evidence in Uncommon Molecularly Defined NSCLC and Other Rare Thoracic Malignancies

Lung cancer represents the most common malignancy in Western countries. In Canada, there were expected to be 32,100 new lung cancer cases in 2024 and 20,700 deaths [36]. Nevertheless, the knowledge of molecular abnormalities (gene mutations or fusions) has grown greatly in the last decade and redefined lung cancer as a series of molecularly defined diseases [37]. Many of these molecularly defined subsets of lung malignancies, such as those with mutations in *HER2*, *MET,* and *BRAF* genes, or fusions in genes such as *ROS1*, *RET*, and *NTRK*, occur in one or two percent of patients with non-squamous NSCLC. These subgroups may occur in as few as 100 or 200 hundred patients nationally in Canada.

Identifying these patients to participate in randomized clinical trials can be very challenging. There may be limited information for understanding if the natural history, response to therapy, and expected survival of these rare molecularly defined cancer subtypes differs from patients with NSCLC and no underlying actionable molecular abnormality. The collection of real-world evidence can inform answers to these questions and potentially open up additional therapeutic options.

There are several examples where data from single-arm clinical trials, supplemented with real-world evidence, have been used in regulatory submissions for drug approval in Canada. *ROS1* fusions are rare gene rearrangements that occur in approximately 1% of patients with NSCLC [38]. There are no randomized clinical trials of therapies in *ROS1*-rearranged NSCLC. Small phase II clinical trials demonstrated high objective response rates (ORRs) and long PFS times for the oral molecularly targeted therapies crizotinib and entrectinib [39,40]. Data from real-world studies support these findings and were important in regulatory and drug reimbursement decisions that resulted in access to effective oral molecularly targeted therapies for patients with *ROS1*-rearranged NSCLC [41,42].

Similarly, thymic epithelial tumors represent a heterogeneous group of rare tumors with little high-quality evidence to inform practice. Clinical practice guidelines for these diseases have adopted a consensus approach, instead of being strongly evidence-informed [43]. Real-world evidence from national databases can help inform the management of these tumors. Databases, such as the French national RYTHMIC database, provide insight into patterns and outcomes of practice for thymic epithelial tumors that help inform future practice or provide direction in research priorities [44].

### 4.2. Real-World Data on Patients with Poor Functional Status or Comorbidity

Lung cancer clinical trials mostly include only patients with a good Eastern Cooperative Oncology Group performance status (ECOG PS 0–1). Additionally, elderly patients, often defined as over 70 years of age, are frequently under-represented in clinical trials. However, patients with ECOG PS 2 and elderly patients are routinely offered systemic therapy in everyday clinical practice. Additionally, patients with thoracic malignancies often have significant associated comorbidities such as chronic obstructive pulmonary disease (COPD), cardiovascular disease, or other comorbidities that might have been exclusion criteria in the pivotal clinical trials establishing the efficacy of a therapeutic intervention. Therefore, concerns have been raised about the generalizability of findings from randomized clinical trials to these populations of patients. Real-world evidence can inform understandings of the prognosis of these patient groups and whether it is justified to extrapolate the randomized clinical findings to them or, alternatively, adopt a different approach to treatment decision-making. It can provide data on the safety and effectiveness of therapy among the populations of patients excluded from the randomized trials, facilitating discussion about the generalizability of trial findings to those populations of patients not included in trial populations.

Analysis of provincial data from the Ontario Institute for Clinical Evaluative Sciences (IC-ES) demonstrated that older patients with lung cancer are less likely to see an oncologist and less likely to receive treatment [45,46]. This may represent referral bias and nihilism about treatment in older lung cancer patients. Analysis of real-world data, though, would suggest that elderly patients receiving an immune-checkpoint inhibitor demonstrate similar efficacy and toxicity to younger patients [47]. Nevertheless, patient selection remains a crucial factor in implementing these findings.

Real-world evidence can also provide cautionary tales. Analysis of real-world data from the Alberta Immunotherapy Database suggests that data from randomized clinical trials cannot be routinely extrapolated to NSCLC patients with poor performance status [48]. Poor performance status (ECOG PS 2 or greater) NSCLC patients treated with an immune-checkpoint inhibitor demonstrated worse overall survival, shorter time to treatment failure, and were more likely to present to emergency departments, be admitted to hospital, and die in hospital during their first admission than patients with good performance status.

### 4.3. Real-World Data to Complement Regulatory Submissions

There is an opportunity for real-world data to complement submissions to regulatory authorities and healthcare funders [23,35]. As stated above, lung cancer has evolved over the last 10–15 years and now consists of a series of subgroups, often defined by underlying molecular abnormalities. However, much of our existing knowledge is derived from trials conducted in unselected populations of patients with NSCLC. As a result, gaps in knowledge exist regarding the natural history of these molecularly defined subgroups of NSCLC, and their responses to existing therapies. The collection of real-world evidence in large databases, such as Flatiron (Flatiron Health, USA), can help address questions, including whether these subgroups of patients derive similar benefits from standard therapies and whether they derive greater benefits from molecularly targeted therapies. This may augment regulatory and reimbursement submissions where there is an absence of randomized clinical trial data.

One recent example of the use of real-world data in drug reimbursement submissions is the Canadian Drug Authority reimbursement review of selpercatinib for *RET*-positive advanced/metastatic NSCLC [49]. The available clinical trial data was a single-arm clinical trial in patients with *RET*-positive metastatic NSCLC. There was a lack of data regarding the efficacy of standard therapies in this molecularly defined subgroup of NSCLC patients. An indirect treatment comparison, including real-world data from the Flatiron database, supported the superior efficacy of selpercatinib in comparison to other standard treatment options for this population of patients, resulting in a positive recommendation for public reimbursement.

### 4.4. Real-World Data as Part of Quality of Care

Real-world data can have an important role in evaluating quality of care. Quality of care research seeks to understand whether care is being delivered appropriately, or whether there are gaps in the implementation and delivery of care. Real-world data can answer questions about patterns of care. An analysis of a consecutive series of patients with small-cell lung cancer (SCLC) seen at the Juravinski Cancer Centre improved understandings of the uptake of prophylactic cranial irradiation at this regional cancer centre, as well as the competing risk of relapse within the central nervous system versus systemic relapse [50]. An analysis of large datasets can provide accurate assessments of the delivery of care. A previous analysis of data from the Ontario Institute of Clinical Evaluative Sciences (IC-ES) demonstrated that elderly patients (aged over 70 years of age) with NSCLC [45] or SCLC [46] were less likely to receive oncologic care. Starting at age 60 or 65, the increasing age of patients with lung cancer in Ontario, Canada, was associated with a lower likelihood of referral to an oncologist and a lower likelihood of the receipt of systemic therapy. These analyses of real-world data identify potential gaps in care and provide opportunities for future research questions aimed at addressing these care-delivery gaps and improving the quality of care.

## 5. Conclusions

Real-world data is increasingly reported in the medical literature. Strengths and limitations exist with such data, as highlighted in the findings of this narrative review. Challenges exist in the methodologies of real-world data collection and, as a result, real-world studies are often subject to significant biases and confounding. Nevertheless, real-world data may complement existing data from randomized clinical trials and play an important role in evaluating patterns and outcomes of care.

## Figures and Tables

**Table 1 curroncol-32-00629-t001:** Strengths of real-world evidence.

Strength	# of Studies Mentioning the Strength
Real-world assessments/assessment of the generalizability of RCT findings (e.g., tolerability, safety, policy, and cost-effectiveness) [10,11,12,13,14,15,16,17,18,19,20,21]	12
Possibility of long-term surveillance [12,13,21,22,23,24,25,26,27,28]	10
Increased study on cohorts/disease states (rare disease) where RCTs are not possible/have not yet been completed. [10,14,17,18,19,21,28,29,30]	9
Resource efficiency [11,12,13,16,19,21,22,25,27,28]	9
Descriptive of treatment patterns under real-world conditions, including under-represented populations [10,13,14,17,18,23,27,28]	8
Increased generalizability [11,12,13,16,17,22,26]	7
Larger sample sizes [10,16,20,23,25,26,27]	7
Researching areas with no/limited data [10,13,14,17,28,29,30]	7
Time-efficiency [11,12,16,19,22,25,27]	7
Increased external validity [12,22,23,25,27]	5
Increased identification of important cohorts for stratification [10,20,21,28]	4
Translation of RCT outcomes into clinical practice [10,19,27,28]	4
Hypothesis-generating [10,19,28]	3
Creation of multi-purpose data collection mechanisms [10,16]	2
Can be generated prospectively and retrospectively [23]	1
Direct-to-patient data collection [23]	1
Lower selection bias [18]	1
Cost of analysis is lower than RCTs [10]	1

## Data Availability

No new data were created or analyzed in this study.

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
