# Peer review of "A Narrative Review of the Strengths and Limitations of Real-World Evidence in Comparison to Randomized Clinical Trials: What Are the Opportunities in Thoracic Oncology for Real-World Evidence to Shine?"

_curroncol, 2025, doi:10.3390/curroncol32110629_

Round 1

Reviewer 1 Report

Comments and Suggestions for Authors

This article addresses the topical issue of real-world data, and discusses the strengths, limitations and relevant opportunities in thoracic cancer research. It provides a systematic summary of existing literature on the topic and offers insights into specific areas of thoracic oncology where real-world data research may be especially useful.

General comments

  1. While the topic of real-world data in oncology research is very topical and relevant to current directions in cancer research, I have reservations that a systematic review is not the best format or article type to address this topic. The purpose of a systematic review is to answer a focused research question by identifying, appraising and synthesising available empirical evidence on the topic. The current publication does not meet those requirements as 1) there is no focused question; 2) the inclusion criteria for article selection are very broad i.e. “they discussed methods to incorporate real world evidence into research, including potential strengths, or limitations of the use of real world evidence, and compared real world evidence to traditionally accepted forms of evidence”; 3) there is no process of appraisal; and 4) the primary sources do not provide empirical evidence, but are generally review articles or editorial-style articles.

A scoping review, narrative review or editorial/opinion piece could be a better format for the contents and message of this article.

  1. Because the topic is not suited for a systematic review, the rationale for undertaking a frequency count of articles that mention each “strength” or “limitation” is tenuous. It is unclear how enumerating the articles that mention an issue adds to the body of knowledge about this topic.
  2. All article selection and data extraction were performed by one author, so there is a significant risk of bias.
  3. The most interesting part of the article is the discussion, which explores aspects of thoracic oncology research that may most benefit from real-world data research. This part warrants expansion and potentially should be made the main focus of the article. This article would be enhanced by examples with citations of how real-world data research has enhanced lung cancer research.
  4. There is a discordance between the contents of this article and the discussion. The inclusion criteria for included articles are not limited to lung cancer, or even cancer research, so all of the summarised strengths and limitations pertain to real-world data research in general. However, the discussion focuses on thoracic oncology examples specifically. Ideally there should be a clearer connection between the methods, results and discussion contents.

Author Response

Comment 1: 

  1. While the topic of real-world data in oncology research is very topical and relevant to current directions in cancer research, I have reservations that a systematic review is not the best format or article type to address this topic. The purpose of a systematic review is to answer a focused research question by identifying, appraising and synthesising available empirical evidence on the topic. The current publication does not meet those requirements as 1) there is no focused question; 2) the inclusion criteria for article selection are very broad i.e. “they discussed methods to incorporate real world evidence into research, including potential strengths, or limitations of the use of real world evidence, and compared real world evidence to traditionally accepted forms of evidence”; 3) there is no process of appraisal; and 4) the primary sources do not provide empirical evidence, but are generally review articles or editorial-style articles.

A scoping review, narrative review or editorial/opinion piece could be a better format for the contents and message of this article.

Response 1

Thank you for the comments. While there is certainly a need for research to systematically summarize non quantitative research, we have removed reference to this being a systematic review and instead will resubmit this as a narrative review. 

The reviewers have questioned what this review is adding to the body of literature. This article was solicited for a special edition on RWE in thoracic oncology. We believe the value of RWE is often overstated in the literature. As such the authors felt there was a strong need for an article discussing the strengths and limitations of RWE within this special edition. 

Changes found  

We have removed any reference to this being a systematic review throughout the article. The methods section has been expended to include the following:

All relevant articles were included, not just those pertaining to oncology, or thoracic oncology, given that the methodologic issues are not oncology specific (materials and methods first paragraph)

Quality assessment was not performed, as the articles retrieved generally did not report on methods, or empiric data and would all be classified as low quality. A tool such as the Cochrane risk of bias[9] could not meaningfully be applied. There were no quantitative data to combine or analyze. (last paragraph methods)

Comment 2

  1. Because the topic is not suited for a systematic review, the rationale for undertaking a frequency count of articles that mention each “strength” or “limitation” is tenuous. It is unclear how enumerating the articles that mention an issue adds to the body of knowledge about this topic

Response 2

We agree that this is a non standard analysis of the data. However, we wanted to provide the reader with some sense of the relative importance of individual strengths and limitations rather than considering them of equal importance, That would simply create an alternate bias. 

A frequency count was used to indicate how often a strength or limitation was raised in the literature, as a surrogate of the relative importance of the issue. These were then ordered in descending frequency (last paragraph methods)

Comment 3

All article selection and data extraction were performed by one author, so there is a significant risk of bias.

Response 3

We agree that having two independent assessors minimizes bias. However, this review was an undergraduate student research project and it was not feasible to have a second reviewer. This cannot be changed at this point, other than redefining this as a narrative review, as described above

Comment 4

The most interesting part of the article is the discussion, which explores aspects of thoracic oncology research that may most benefit from real-world data research. This part warrants expansion and potentially should be made the main focus of the article. This article would be enhanced by examples with citations of how real-world data research has enhanced lung cancer research.

Response 4

Thank you for your thoughts on this. We agree that understanding the place of RWE is the more important part of this article. We have made multiple changes to this section and expanded on examples of RWE in thoracic oncology

Comment 5

There is a discordance between the contents of this article and the discussion. The inclusion criteria for included articles are not limited to lung cancer, or even cancer research, so all of the summarised strengths and limitations pertain to real-world data research in general. However, the discussion focuses on thoracic oncology examples specifically. Ideally there should be a clearer connection between the methods, results and discussion contents.

Response 5. 

As noted above in response 1, we added a comment in the methods to expand on why the review covers more than just oncology literature. There is little literature in thoracic oncology regarding this issue. The methodologic issues extend beyond oncology and so all relevant literature was included in the review. However, the focus of the discussion is in the area of thoracic oncology, as the article is written for a special edition on RWE in thoracic oncology. 

All relevant articles were included, not just those pertaining to oncology, or thoracic oncology, given that the methodologic issues are not oncology specific (materials and methods first paragraph)

Reviewer 2 Report

Comments and Suggestions for Authors

The research question is clear: to compare the strengths and limitations of real-world evidence (RWE) with randomized clinical trials (RCTs). However, the paper does not provide a truly novel theoretical or empirical contribution. Most of the identified content (e.g., strengths such as generalizability and limitations such as bias and confounding) is already well known in the RWE vs. RCT debate. This review largely reiterates concepts that are already well established (internal versus external validity, bias in RWE, lack of standardization) without offering new insights. Although the conclusions are correct in emphasizing the complementary, but not substitutive, role of RWE in relation to RCTs, they remain rather generic. The section on thoracic oncology is potentially valuable, but at present it is mostly descriptive and would benefit from stronger data support or more concrete examples.

Major Concerns

1) Methodology: some key methodological steps are missing. The literature search was limited to only two databases (Medline and Healthstar), which is insufficient to ensure comprehensive coverage. Additional databases should have been included.

2) Reviewer process: Title/abstract screening, full-text review, and data extraction were conducted by a single author. According to guidances, at least two independent reviewers are required to minimize selection bias and errors.

3) Lack of assessment: although the review summarizes strengths and limitations of real-world evidence, it does not formally assess the methodological quality of the included papers. Without a structured risk of bias evaluation, the reliability of the findings remains uncertain.

Author Response

Comment 1

Methodology: some key methodological steps are missing. The literature search was limited to only two databases (Medline and Healthstar), which is insufficient to ensure comprehensive coverage. Additional databases should have been included.

Response 1

Thank you for your feedback. Methodologic issues were raised by reviewer 1 as well.  In response to this we have removed reference to this being a systematic review and instead will resubmit this as a narrative review. This review was an undergraduate student research project. The student did consult with a librarian prior to the literature search and used the recommended databases. Nevertheless, this point is also addressed as we are resubmitting this as a narrative review.

The reviewers have questioned what this review is adding to the body of literature. This article was solicited for a special edition on RWE in thoracic oncology. We believe the value of RWE is often overstated in the literature. As such the authors felt there was a strong need for an article discussing the strengths and limitations of RWE within this special edition, in combination with an opinion piece about the role of RWE in thoracic oncology. 

Comment 2

Reviewer process: Title/abstract screening, full-text review, and data extraction were conducted by a single author. According to guidances, at least two independent reviewers are required to minimize selection bias and errors. 

Response 2

We agree that having two independent assessors minimizes bias. This point was also raised by another reviewer, However, this review was an undergraduate student research project and it was not feasible to have a second reviewer. This cannot be changed at this point, other than redefining this as a narrative review, as described above

Comment 3

Lack of assessment: although the review summarizes strengths and limitations of real-world evidence, it does not formally assess the methodological quality of the included papers. Without a structured risk of bias evaluation, the reliability of the findings remains uncertain.

Response 3

Again, we are resubmitting this as a narrative review. Nevertheless, the following comment has been added into the methods.

Quality assessment was not performed, as the articles retrieved generally did not report on methods, or empiric data and would all be classified as low quality. A tool such as the Cochrane risk of bias[9] could not meaningfully be applied. There were no quantitative data to combine or analyze. (last paragraph methods)

Reviewer 3 Report

Comments and Suggestions for Authors

Review report on current oncology-3866134 manuscript entitled ‘A systematic review of the strengths and limitations of real world evidence in comparison to randomized clinical trials: what are the opportunities in thoracic oncology for real world evidence to shine?

This study reviewed the strengths and limitations of 22 real-world evidence studies and suggested the use of real-world evidence as a complement to randomized controlled trials.

Overall, the research method, discussion, and authors’ proposals appear reasonable. The key message of this study seems to be the four applications of the real-world study that the authors proposed in the discussion.

Please specify the titles of these four applications.

For example, in the case of ‘real world evidence in rare disease’. This disease is limited to thoracic cancers. Therefore, reflecting the authors’ description, it would be better to refer to it as a rare subgroup in molecular profiling rather than a ‘rare disease’. Similarly, ‘real world data as part of quality assurance’ would be better expressed as ‘quality of care’

A few other typos need to be corrected.

The hyphenation of ‘non small cell lung cancer’ in the first paragraph on page 4, ‘astute’ on page 5, and ‘cognizant’ on page 6 are suspected typos.     

Author Response

Comment 1

This study reviewed the strengths and limitations of 22 real-world evidence studies and suggested the use of real-world evidence as a complement to randomized controlled trials.

Overall, the research method, discussion, and authors’ proposals appear reasonable. The key message of this study seems to be the four applications of the real-world study that the authors proposed in the discussion.

Response 1

Thank you for your positive feedback. 

Comment 2

Please specify the titles of these four applications.

For example, in the case of ‘real world evidence in rare disease’. This disease is limited to thoracic cancers. Therefore, reflecting the authors’ description, it would be better to refer to it as a rare subgroup in molecular profiling rather than a ‘rare disease’. Similarly, ‘real world data as part of quality assurance’ would be better expressed as ‘quality of care’

Response 2

section 4.1 has been relabelled 

4.1. Real world evidence in uncommon molecularly defined NSCLC and other rare thoracic malignancies

section 4.4 has been relabelled

4.4. Real world data as part of quality of care

4.2 and 4.3 remain the same

Comment 3

A few other typos need to be corrected.

The hyphenation of ‘non small cell lung cancer’ in the first paragraph on page 4, ‘astute’ on page 5, and ‘cognizant’ on page 6 are suspected typos.     

Response 3

non small cell lung cancer has been changed to non-small cell lung cancer

Cognizant is spelled correctly, as is astute so I am not sure what the reviewer was requesting

Round 2

Reviewer 1 Report

Comments and Suggestions for Authors

Thank you for addressing my comments and updating the format of the review. I think the discussion has been enriched by the specific examples of RWD research in thoracic oncology.

Author Response

Comment 1

Thank you for addressing my comments and updating the format of the review. I think the discussion has been enriched by the specific examples of RWD research in thoracic oncology.

Response 1

Thank you for your feedback and I am pleased you feel everything was addressed

Reviewer 2 Report

Comments and Suggestions for Authors

In revising the manuscript, authors have reclassified it as a narrative review following earlier concerns about missing key methodological steps for a systematic review. However, the current draft still largely retains the structure and style of a systematic review. A narrative review is not simply a summary of studies; its value lies in critically synthesising the literature, offering interpretation, and providing a clear conceptual perspective. To strengthen your work, I encourage authors to move beyond cataloguing findings and instead highlight your own critical appraisal, integrate insights across studies, and articulate how your synthesis advances the conversation in thoracic oncology.

  1. Clarity of Title: For transparency, it would be advisable to explicitly state “A Narrative Review” in the title.

  2. Abstract: The abstract is informative but reads more like that of a structured review. Please emphasise the interpretative stance of the authors and clarify that this is a narrative review.

  3. Discussion: A narrative review should reflect the authors’ perspective, offering a conceptual framework or advancing the debate. At present, the discussion mainly reports what was found, with limited critical synthesis. I encourage you to make the discussion less of a summary of findings and more focused on the authors’ interpretation and critical appraisal, highlighting your own perspective on the implications of the reviewed literature.

  4. Discussion: While you highlight concerns about bias and confounding, the manuscript would benefit from a brief acknowledgement of how these issues might be addressed in real-world studies, for example, through quasi-experimental approaches such as propensity score methods. A very short mention of these methods would enrich the discussion.

Author Response

Comment 1

In revising the manuscript, authors have reclassified it as a narrative review following earlier concerns about missing key methodological steps for a systematic review. However, the current draft still largely retains the structure and style of a systematic review. A narrative review is not simply a summary of studies; its value lies in critically synthesising the literature, offering interpretation, and providing a clear conceptual perspective. To strengthen your work, I encourage authors to move beyond cataloguing findings and instead highlight your own critical appraisal, integrate insights across studies, and articulate how your synthesis advances the conversation in thoracic oncology.

  1. Clarity of Title: For transparency, it would be advisable to explicitly state “A Narrative Review”in the title.

Response 1

As requested the title has been modified to include ‘A narrative review …’

Comment 2

  1. Abstract: The abstract is informative but reads more like that of a structured review. Please emphasise the interpretative stance of the authors and clarify that this is a narrative review.

Response 2

We have included the term ‘narrative review/ and reordered the abstract for clarity. However, we would disagree with the reviewer on this comment. There is a limited amount of space in the abstract. The comments in the abstract are not repetitive of the review, but are interpretative to the field of oncology and offer insights into the role of real world evidence in oncology.

Comment 3

  1. Discussion: A narrative review should reflect the authors’ perspective, offering a conceptual framework or advancing the debate. At present, the discussion mainly reports what was found, with limited critical synthesis. I encourage you to make the discussion less of a summary of findings and more focused on the authors’ interpretation and critical appraisal, highlighting your own perspective on the implications of the reviewed literature.

Response 3

We do not entirely agree with this comment. As stated in the initial reviewer’s comments, the articles included in this review do not have empiric data. Providing a conceptual framework is beyond the scope of this article. Within the scope of this special edition on real world evidence in thoracic oncology, the purpose of this paper is to provide a commentary about the role of real world evidence in thoracic oncology. The review was to provide a background for this, that is unlikely to be contained in other papers in the special edition. There are significant limitations to real world evidence in oncology and therefore there should be defined roles for real world evidence to be complimentary to high quality evidence from randomized clinical trials.

I am not sure that the reviewer has appreciated where we have already added out thoughts in the presentation of the results. Nevertheless, we have added more detail to the section entitled results, in order ton convey our thoughts more fully. We believe the critical thinking component has well covered in the section discussing the role of real world evidence in thoracic oncology

Comment 4

  1. Discussion: While you highlight concerns about bias and confounding, the manuscript would benefit from a brief acknowledgement of how these issues might be addressed in real-world studies, for example, through quasi-experimental approaches such as propensity score methods. A very short mention of these methods would enrich the discussion.

Response 4

We have added some information into the discussion to address these points

Round 3

Reviewer 2 Report

Comments and Suggestions for Authors

No comments